behaviour, neuroscience, ecology

oestrogen receptor, prairie vole, RNAi, social monogamy

**Author for correspondence:**
Connor T. Lambert
e-mail: lamberc2@miamioh.edu

†Co-senior authors.

# Medial amygdala ERα expression influences monogamous behaviour of male prairie voles in the field

Connor T. Lambert[1], James B. Lichter[1], Adam N. Perry[3], Samuel A. Castillo[3], Brian Keane[2,†], Bruce S. Cushing[3,†] and Nancy G. Solomon[1,†]

[1]Department of Biology, Miami University, Oxford, OH 45056, USA
[2]Department of Biological Sciences, Miami University—Regionals, Hamilton, OH 45011, USA
[3]Department of Biological Sciences, The University of Texas at El Paso, El Paso, TX 79968, USA

CTL, 0000-0002-3568-1868; BSC, 0000-0002-8676-0179

Formation of long-term pair-bonds is a complex process, involving multiple neural circuits and is context- and experience-dependent. While laboratory studies using prairie voles have identified the involvement of several neural mechanisms, efforts to translate these findings into predictable field outcomes have been inconsistent at best. Here we test the hypothesis that inhibition of oestrogen receptor alpha (ERα) in the medial amygdala of male prairie voles would significantly increase the expression of social monogamy in the field. Prairie vole populations of equal sex ratio were established in outdoor enclosures with males bred for high levels of ERα expression and low levels of prosocial behaviour associated with social monogamy. Medial amygdala ERα expression was knocked down in half the males per population. Knockdown males displayed a greater degree of social monogamy in five of the eight behavioural indices assessed. This study demonstrates the robust nature of ERα in playing a critical role in the expression of male social monogamy in a field setting.

## 1. Introduction

Even though social monogamy in mammals is relatively rare, estimates ranging from 5% to 15% [1,2], it has been a highly studied phenomenon. Social monogamy is classified by a suite of prosocial behaviours, which includes the formation of long-term male–female pairs bonds and bi-parental care of offspring, rather than mating exclusivity (genetic monogamy). Although rare, social monogamy has evolved in almost all major mammalian taxa, with estimates that it has evolved independently at least 61 times [1]. It has not only evolved between taxa but also repeatedly within a mammalian taxon, for example, it is hypothesized to have evolved at least 20 times in rodents [1] and six times within primates [3]. The repeated evolution of social monogamy has generated substantial investigation into the selective forces involved in the evolution of social monogamy from solitary breeding species, especially in terms of monogamous behaviours in males [1]. It has been argued that since male mammals cannot contribute significantly to gestation or providing nutrition for preweanling offspring, monogamy should only be selected for if it favours male fitness relative to other mating systems. Since male mammals do not typically contribute heavily to rearing offspring, male parental care involves significant shifts in male behaviour [4]. Hypotheses for why monogamy evolves have included potential ecological explanations, such as conditions that limit resources and population density, thereby limiting a male's ability to access multiple females [5].

From the perspective of behavioural neuroscience, social monogamy has provided significant insight into the neural mechanisms regulating high

levels of prosocial behaviour. It is also of interest because while there may be a limited number of neural mechanisms and neural circuits involved (see below), selection may have acted differentially on the same mechanisms in reorganizing the brain, creating convergent social behaviour patterns [6]. This means that in addition to understanding proximate mechanisms involved in regulating high levels of prosocial behaviour, species such as the prairie vole (*Microtus ochrogaster*), have become important relevant translational model systems for understanding neural mechanisms associated with social deficit disorders and drug abuse in humans [7]. While laboratory studies have been invaluable in elucidating neural mechanisms and circuits involved in regulating social monogamy and prosocial behaviour, the next critical step is to examine if these highly controlled experimental results translate into predictable outcomes under the greater complexity of natural conditions. Therefore, the goal of this study was to test the hypothesis that oestrogen receptor alpha (ERα) would directly translate into an expression of social monogamy under field conditions.

Studies focusing on the prairie model system have provided a number of significant insights into the neural regulation of prosocial behaviour, including the following. (i) There are at least two major neural networks involved, social and reward, as well as other regions/nuclei which play a critical role in modulating behavioural responses [8]. (ii) Multiple neural mechanisms have been shown to influence/regulate social monogamy, including oxytocin [9,10], vasopressin [11], dopamine [12,13], corticosterone [14] and oestrogen receptor expression [15]. (iii) The patterns of receptor expression for oxytocin [16], vasopressin [17], oestrogen [18], dopamine [19] and corticotrophin-releasing factor [20] vary predictably with reproductive strategies, degree of prosocial behaviour and aggression. While there are multiple factors that have been shown to influence the expression of male behaviours associated with social monogamy, much of the research has focused on a single factor, arginine vasopressin (AVP), specifically the expression of vasopressin 1a receptors (V1aR) and the *avrp1a* microsatellite region associated with the gene encoding V1aR [21–25]. Although laboratory studies have indicated a major influence of AVP, attempts to translate these studies to other socially monogamous mammals [26] or field settings have either not produced the predicted results or produced conflicting results [27–33]. While several studies have found a correlation between male *avpr1a* genotype and the number of females they sire offspring with, they have not shown direct relationships with either genetic or social monogamy [28,29,32,33]. Interestingly, one study by Keane *et al.* [33] did find a relationship between the number of females with which a male sires offspring and *avpr1a* microsatellite length, but it was the opposite of the predicted relationship with males with longer microsatellites actually breeding with more females, while others found no relationships between *avpr1a* genotype and any aspect of the mating system [27,29,30,34]. The results of these studies clearly indicate that other factors are playing a critical role in behaviours linked to monogamy [26,35,36].

The lack of translation from laboratory effects of oxytocin and vasopressin to prosocial behaviour in the field may not be surprising, because the formation of pair bonds is one of the most complex social behaviours and can be influenced by numerous intrinsic (e.g. social experience affects oxytocin expression [37]) and extrinsic factors, such as environmental conditions (i.e. vasopressin plays a major role in water balance [38]), population density, resource distribution and/or their interactions. Here, we predict that ERα expression will be a reliable predictor of the expression of social monogamy under ecologically relevant conditions.

It has been hypothesized that the expression of social monogamy is an interaction between steroids and neuropeptides [35], supported by the fact that many of the behavioural effects of the nonapeptides vasopressin and oxytocin are steroid dependent. This is supported by the fact that social recognition, an essential aspect of pair-bond formation, is the product of an oestrogen-dependent four-gene interaction, consisting of vasopressin, oxytocin and oestrogen receptors [39]. In male mice, it has recently been hypothesized that social disorders in which individuals lack the ability to form bonds or display preferences, such as autism and schizophrenia, are caused by an interaction between oestrogen receptors and gene expression of vasopressin and oxytocin receptors [40]. Additionally, comparison of independent studies of the effects of gonadal steroids and vasopressin/oxytocin indicate that they regulate many of the same social behaviours (for reviews, see [35,39,40]).

In prairie voles, ERα expression in critical brain regions has been shown to be necessary and sufficient for the expression of high levels of male prosocial behaviour [15,41]. Low levels of ERα in the medial amygdala (MeA) and bed nucleus of the stria terminals (BST), critical brain regions in the expression of sociosexual behaviour, are directly linked to pair-bonding and parental care in male arvicoline rodents [18,42]. By contrast, enhancing MeA or BST ERα disrupts partner preferences and alloparental behaviour in male prairie voles [15,43], while decreasing MeA ERα in polygynous male meadow voles (*Microtus pennsylvanicus*) increases prosocial behaviour and decreases inter-male aggression [41]. Other species display distinct differences in social behaviour linked to MeA ERα as well; ERα expression in the social brain network is connected to the mating systems of *Peromyscus* species [6], and white-throated sparrows (*Zonotrichia albicollis*) have markedly different territorial and mating behaviours associated with MeA ERα differences [44,45]. The relationship between regional ERα expression and prosocial behaviour not only occurs among other species including *Microtus* species [18,42] but is also expressed between geographically and culturally distinct populations of prairie voles. Male prairie voles from Kansas (KS) display significantly lower levels of prosocial behaviour and higher levels of aggression than males from Illinois (IL) and KS males express higher levels of ERα in the MeA (figure 1) and BST than their IL counterparts [47]. These differences are further exaggerated in $F_1$ male offspring from KS dams and IL sires (KI) that show the lowest level of male prosocial behaviour [48] and significantly higher levels of ERα in the MeA than either KS or IL males [46].

Here we test the prediction that KI males with MeA ERα knocked down will display higher levels of social monogamy in a field setting than control KI males. We tested this prediction by establishing eight replicate semi-natural populations in 0.1 ha outdoor enclosures, each consisting of eight adult IL females and eight adult KI males (four MeA-ERα knocked down males and four luciferase-short hairpin RNA (shRNA) control males). All founding individuals were radio-collared and monitored for 15 weeks. After that time, all animals

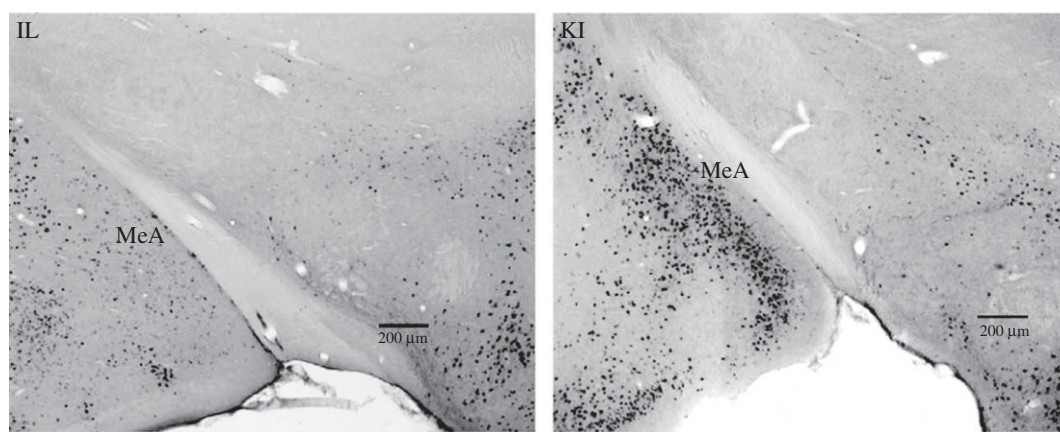

**Figure 1.** Photomicrograph showing the overexpression of ERα in the medial amygdala (MeA) of KI males compared with IL males. Data and image originally appeared in *Neuroscience* in 2006 [46].

**Table 1.** Summary of indices of social and genetic monogamy.

| index | data used | weeks | details |
|---|---|---|---|
| **nest residency and associations** | | | |
| residency | nest-trapping and radio nest-checks | 6–8,9–11,12–14 | resident: more than 75% of detections at one nest |
| social monogamy | nest-trapping and radio nest-checks | 6–8,9–11,12–14 | male resident at nest with only 1 female resident |
| relative AI | nest-trapping and radio nest-checks | 6–8,9–11,12–14 | max(AI)/sum(all AIs) |
| **space-use (grid-trapping)** | | | |
| max overlap | grid-trapping | 8,11,14–15 | maximum capture sites shared w/1 female |
| proportion overlapped | grid-trapping | 8,11,14–15 | proportion of females in enclosure sharing a capture site w/male |
| **space-use (radio-tracking)** | | | |
| home range size | radio-tracking | 3–6,7–10,11–14 | area (m$^2$) of 75% kernel utilization distribution |
| max overlap | radio-tracking | 3–6,7–10,11–14 | maximum overlap of male's home range by a female |
| proportion overlapped | radio-tracking | 3–6,7–10,11–14 | proportion of females in an enclosure overlapped by male's home range |
| adult survival | radio-tracking | 1–15 | weeks survived |
| **genetic indices** | | | |
| genetic monogamy | parentage data | 1–5,6–10,11–15 | genetically monogamous: only 1 female mated in a period |
| offspring sired | parentage data | 1–15 | total no. offspring sired |
| litters sired | parentage data and trapping data | 1–15 | total no. litters sired |
| offspring survival | parentage data and trapping data | 1–15 | estimated no. days survived |

were trapped from enclosures and knockdown effectiveness determined in founder males that survived until the end of the experiment. Radio-tracking and live-trapping data were used to generate eight behavioural indices associated with social monogamy, including nest residency, associations between males and females and space use measured by home range size and overlap (table 1).

## 2. Results

### (a) Neuroanatomical

Of the 32 RNAi transfected males, 15 survived to the end of the study and their brains were collected and analysed for MeA RNAi hits. Based upon green fluorescent protein (GFP) (figure 2) staining of the 15 RNAi, nine were confirmed MeA hits; five expressed GFP, but not in the MeA and in one

male no staining was detected. The five males with no MeA staining displayed GFP detected in other brain regions that included the central amygdala, supraoptic nucleus and ventroposteriolateral thalamus. Examination of the effects of RNAi treatment using RNAscope (HiPlex ADC) revealed a complete knockdown of ERα mRNA in the regions that expressed GFP confirming the GFP results.

### (b) Field

Two analyses were conducted, one comparing data from only confirmed MeA-ERα knockdown males ($n = 9$; figure 2d) against control males ($n = 29$) and a more conservative comparison of data from all ERα RNAi-transfected founding males ($n = 32$), regardless of whether MeA-ERα knockdown could be confirmed, against control males. The analyses presented first include only the surviving males in which ERα

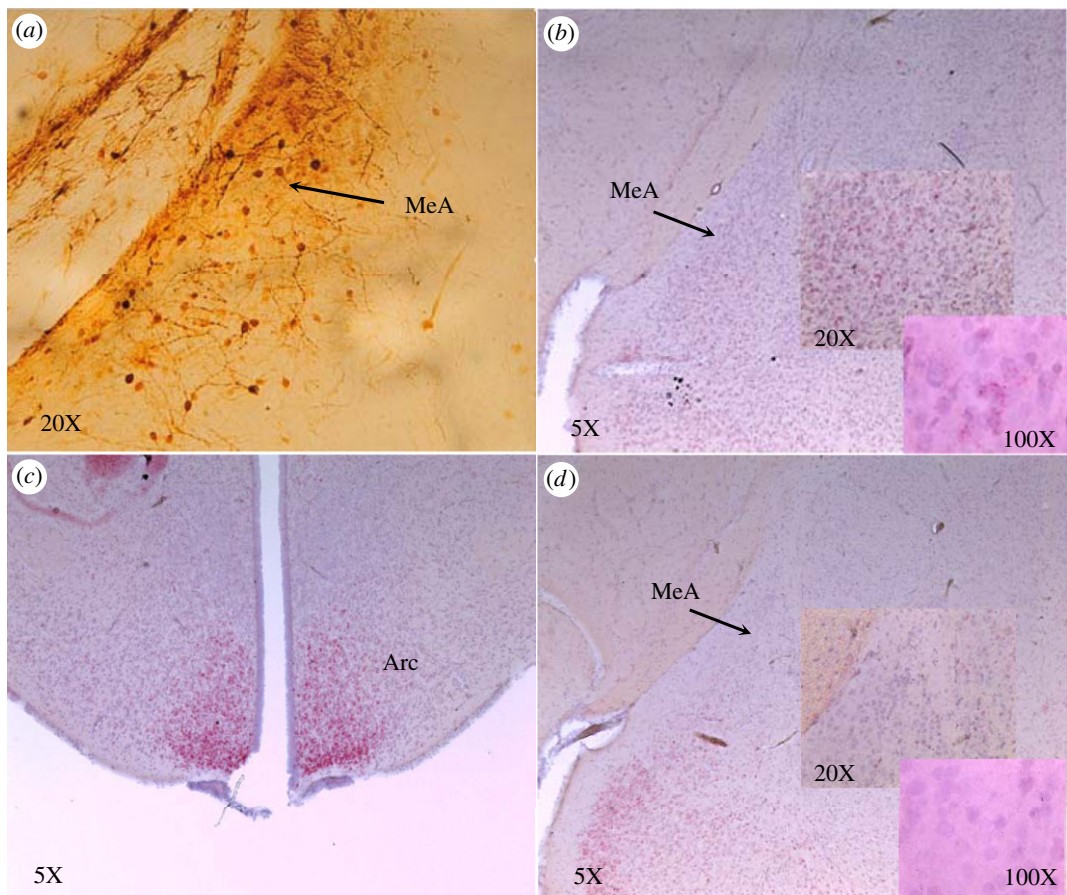

**Figure 2.** Photomicrograph showing ERα-RNAi transfection in the medial amygdala (MeA) and ERα mRNA expression in a knockdown and control male. (*a*) Green fluorescent protein (GFP) immunoreactivity expression in the MeA of an ERα-RNAi transfected KI male. GFP is visualized using immunohistochemistry and diaminobenzidine. (*b*) Expression of ERα mRNA in the medial amygdala (MeA) of control male visualized using RNAscope. (*c*) Expression of ERα mRNA in successfully transfected male. (*d*) The absence of ERα mRNA in the MeA of the same ERα-RNAi transfected male. All images are from males collected at the end of the 15-week field study. (Online version in colour.)

inhibition in the MeA was confirmed. As predicted, MeA-ERα RNAi males displayed significantly higher levels of social monogamy than control males (figure 3). MeA-ERα RNAi males displayed higher rates of monogamous residency, a higher relative association index (AI) with one specific female (see Methods), smaller home ranges, a greater proportion of home range overlap with only one female and overlap with fewer females than control males. Social monogamy is based upon a suite of social behaviours and is not necessarily linked with genetic monogamy, which is supported by our finding that there was no difference between the proportion of MeA ERα knocked down (4/8) and control (8/20) males that bred with only a single female (Fisher's exact test $p = 0.69$). There was also no relationship between the degree of socially monogamous behaviour and reproductive success because there was no significant difference in a mean number of total offspring sired (mean ± s.e. control males = 3.45 ± 0.71, MeA-ERα knockdown males = 4.67 ± 0.96, $\chi_1^2 = 0.002$, $p = 0.97$) or the survival of offspring sired by knockdown (37.69% ± 2.86) versus control sires (44.01% ± 1.80) ($\chi_1^2 = 0.04$, $p = 0.84$).

To determine the robustness of the results, we also analysed the data by comparing all ERα RNAi-transfected founding males ($n = 32$), regardless of whether MeA-ERα knockdown could be confirmed, against control males (figure 4). These results showed that six of the eight indices produced the same findings as the main comparison using only surviving confirmed MeA-ERα knockdown males

(figure 3). The main difference was that there was a shift in two of the indexes. When all males were compared there was no longer a significant difference in monogamous residency (Fisher's exact test $p = 0.67$), while there was a significant difference in the maximum proportion of capture sites shared with a female ($\chi_3^2 = 9.54$, $p = 0.02$). The groups again did not differ in rates of genetic monogamy (not shown in figure 4).

Inclusion of all ERα RNAi-transfected founding males did not alter the finding that there was no significant difference in the likelihood of being a resident ($\chi_1^2 = 0.07$, $p = 0.80$) or proportion of females within an enclosure with which males shared space ($\chi_1^2 = 0.07$, $p = 0.79$). Nor did inclusion of all males change the previously reported significant effect in the relative AI ($\chi_1^2 = 6.31$, $p = 0.01$), home range size ($\chi_1^2 = 4.90$, $p = 0.03$), maximum proportion of a male's home range overlapped by one female (all males: $\chi_1^2 = 9.28$, $p = 0.002$) or the number of females overlapped (all males: $\chi_1^2 = 4.64$, $p = 0.03$) between ERα RNAi transfected and KI control males.

Additional tables containing model results are available in the electronic supplementary material.

## 3. Discussion

This is the first study in which a mechanism that has been shown to be critical in the regulation of social monogamy in laboratory studies (reduced ERα expression in the MeA) has

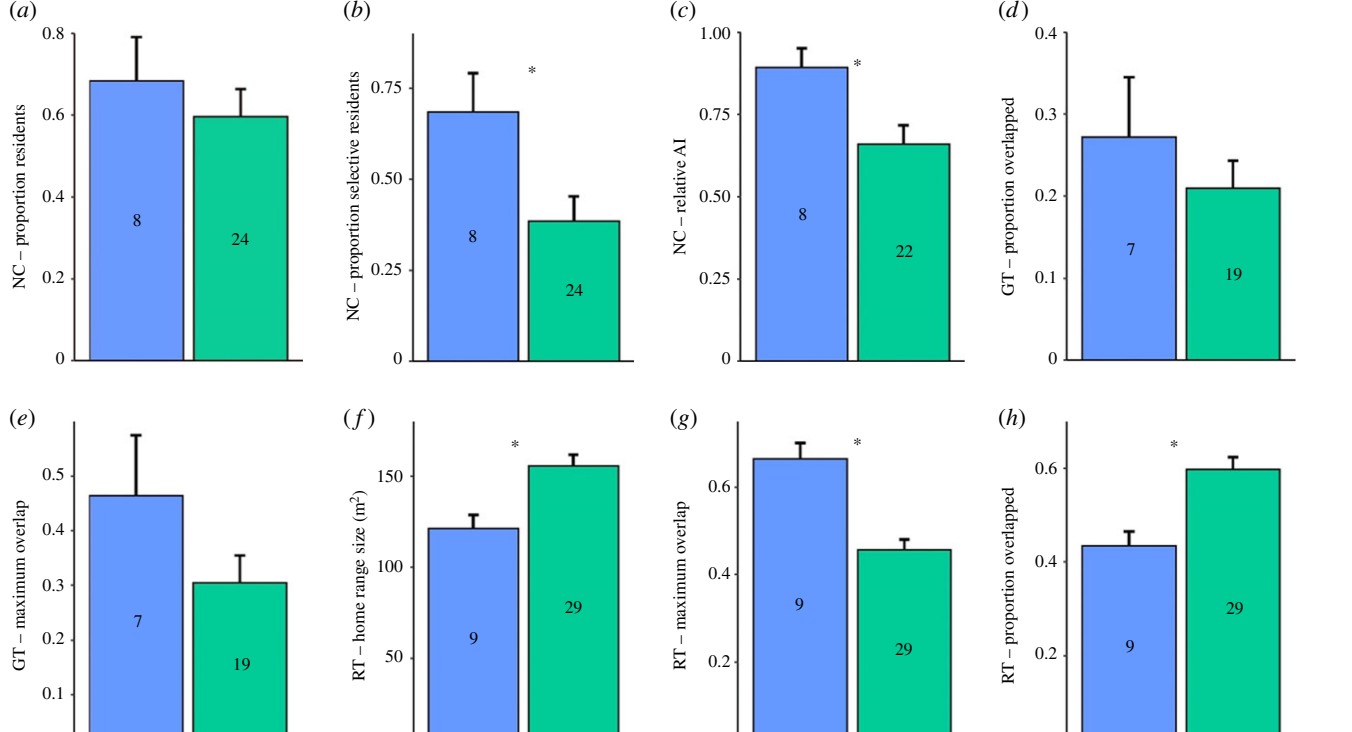

**Figure 3.** Results comparing the eight indices of social monogamy between verified MeA-ERα knock down (blue/left) and control KI males (green/right). Indices from nest-checking (a–c). Indices from grid trapping (d and e). Indices from radio-tracking data (f–h). Relative AI = relative association index, a metric of how strongly a male is associated with one female compared to all female associates. Maximum overlap = the maximum spatial overlap a male has with all females he overlaps. Proportion overlapped = proportion of females in the enclosure with which a male overlaps. * = significant difference between treatments ($p < 0.05$). (Online version in colour.)

also produced consistent and predicted results when manipulated in an ecologically relevant environment. The robust effects following the manipulation of ERα expression contrast with the consistent lack of direct translation associated with other mechanisms (e.g. *avpr1a* microsatellite length) to predictable outcomes in field studies [27,29,30,33]. The difficulty observing critical effects of neuropeptides on prosocial behaviour in the field may not be surprising, because the formation of pair bonds is one of the most complex social behaviors, regulated/influenced by multiple mechanisms [35] within multiple neural circuits [8]. Under natural conditions, many factors including experience (early social experience affects oxytocin expression [37]), environmental conditions (e.g. vasopressin plays a major role in water balance) and/or population density may affect neural responses and/or the underlying mechanisms individually or in concert. However, ERα expression in the MeA may be a more reliable determinant of prosocial behaviour under a wider array of contexts, because the MeA is one of the initial processing stations for social olfactory cues. ERα expression is largely determined before adulthood [49] and only modestly affected by subsequent social manipulations [50]. In turn, the efferent connections of the MeA regulate other nuclei and regions within both the social and reward neural circuits, which are essential in the formation of pair bonds [15]. Therefore, although the MeA may not directly regulate the behaviours analysed, it is likely a keystone region in which manipulation of receptors could have a multifaceted and reliable impact on downstream responses.

Our results are consistent with laboratory findings where MeA-ERα enhancement in male prairie voles inhibited the formation of partner preferences and alloparental behaviour [15], while knocking down ERα in male meadow voles increased male prosocial behaviour [41]. ERα in the BST also plays a critical role in male prairie vole prosocial behaviour [43]. Therefore, these field results would be predicted to be even stronger if ERα had been concurrently reduced in the BST.

Several aspects of our findings suggest that the effects are robust. First, in our analyses that included only the surviving males in which ERα was verified to be successfully knocked down, all but one of our indices of social monogamy were in the direction predicted. Five out of these eight social monogamy indices significantly differed, and the one non-significant social monogamy index that was in the opposite direction was from our grid-trapping data, which had the fewest data points and thus also the smallest sample size. Second, a more conservative analysis using all ERα RNAi-transfected males ($n = 32$), including ones that did not survive until the end of the study preventing knockdown confirmation, still produced a number of highly significant results, which were similar to the results using only the subset of males in which ERα knockdown was confirmed. This suggests that changes in behaviour are so robust that the inclusion of potential ERα knockdown failures, which would be predicted to respond as control males, did not overly weaken the findings. The current findings not only support previous laboratory-based results [15], they also demonstrate the critical role of ERα in the expression of prosocial behaviour under the complex conditions found in ecologically relevant field conditions.

## 4. Methods

### (a) Animals

All animals used for this experiment were housed and bred at the Miami University Animal Care Facility. All procedures were

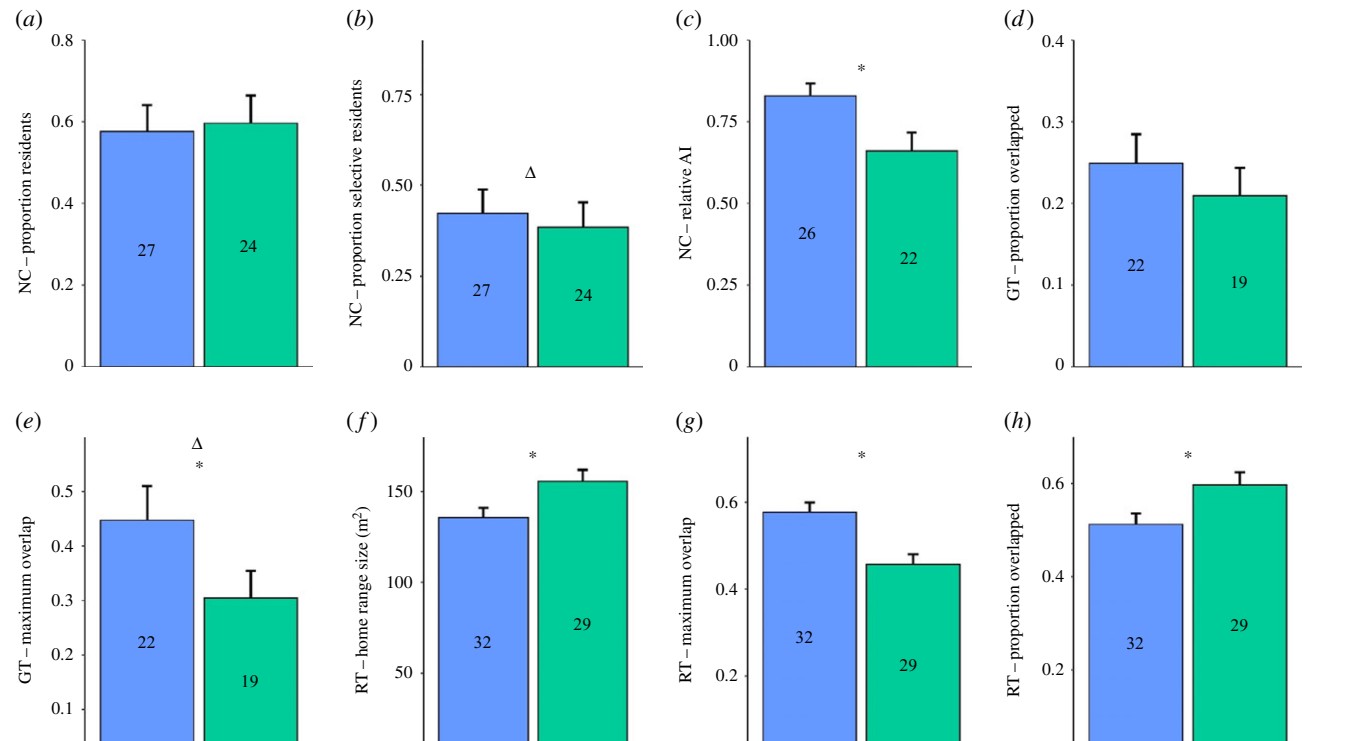

**Figure 4.** Results comparing the indices of social monogamy between MeA-ERα knock down (blue/left) and control KI males (green/right). Indices from nest-checking (*a*–*c*). Indices from grid trapping (*d,e*). Indices from radio-tracking data (*f*–*h*). Relative AI = relative association index, a metric of how strongly a male is associated with one female compared to all female associates. Maximum overlap = the maximum spatial overlap a male has with all females he overlaps. Proportion overlapped = proportion of females in the enclosure with which a male overlaps. * = significant difference between treatments ($p < 0.05$). Δ = change in significance of the results in comparison to the analysis including only males with verified RNAi hits (figure 3). (Online version in colour.)

approved by the Miami University IACUC and followed guidelines for using wild mammals in research from the American Society of Mammalogists and in accordance with the National Institutes Health Guide for the Care and Use of Laboratory Animals.

## (b) Field site

The eight 0.1 ha enclosures that were used to house the semi-natural vole populations were located at the Miami University Ecology Research Center (ERC; 39°53′ N, 84°73′ W), Oxford, Ohio. The vegetation within enclosures consisted primarily of perennial grasses and forbs, which provided food and cover for voles [51].

## (c) Treatments

Founding males were the offspring of a Kansas dam and an Illinois sire (KI), which have higher levels of ERα in the MeA than either Kansas or Illinois males [15,46]. At least two weeks prior to placement in the outdoor enclosures, all males had their MeA transfected using an adeno-associated viral vector, either AAV-H1.ER1 (generously provided by Sergei Musatov) to knockdown ERα (MeA-ERα RNAi) or AAV.H1.Luc containing an anti-luciferase shRNA (controls). The AAV-H1.ER1 vector has been shown to silence/knockdown ERα expression in microtines [41]. Both vectors express a green fluorescent protein reporter from an IRES. We released sixteen voles into each of the eight enclosures: four MeA-ERα RNAi males, four control males and eight females. All founding females were from IL and all founders were released into enclosures on the same day. Prior to release, founders were fitted with a radiocollar (model PD-2C; Holohil Systems Ltd., Carp, Ontario, Canada), uniquely marked, and had a tissue sample taken prior to release for subsequent genetic analysis.

## (d) Radio-tracking

We conducted radio-tracking to determine home ranges of all voles and locate females' nests. Beginning on the day after release, all founding voles were radio-tracked four times per week, twice in the morning and twice in the afternoon, throughout the 15-week study using a hand-held receiver (Advanced Telemetry Systems, Isanti, MN, USA) and Yagi antenna (AF Antronics, Inc., Urbana, IL, USA). Their locations were recorded via Trimble Geo 7x GPS (Trimble Navigation Limited, Sunnyvale, CA, USA) which is accurate to within 1 m. Females' nests were located starting in week two and then checked once per day Monday–Friday until the end of the study.

## (e) Live-trapping procedures

The first pups born in enclosures should have not appeared above ground until about the sixth week of the study (21 days gestation, 21 days until weaning; [52]), therefore we began live-trapping six weeks into the study using 240 × 60 × 90 mm Ugglan multiple-capture traps (Grahnab, Sweden), and alternated between trapping at nests and in a grid. Each trap was covered with an aluminium shield and vegetation to protect animals in traps from heat and precipitation. Traps were baited with cracked corn, a low-quality food [53]. We trapped at nests during weeks 6–7, 9–10 and 12–13 to capture voles visiting or residing at the nest, capture offspring born at the nest, and determine nest residency, social monogamy and the relative AI (detailed below). During each nest trapping period, three Ugglan traps were placed within 1 m of entrance(s) of nests, and we checked traps 10 times each week, five mornings and five evenings.

We also trapped in a grid during weeks 8, 11, 14 and 15, which allowed us to determine the females with which a male shared space (detailed below) and catch voles that were not trapped at a specific nest. For grid-trapping, we placed 25

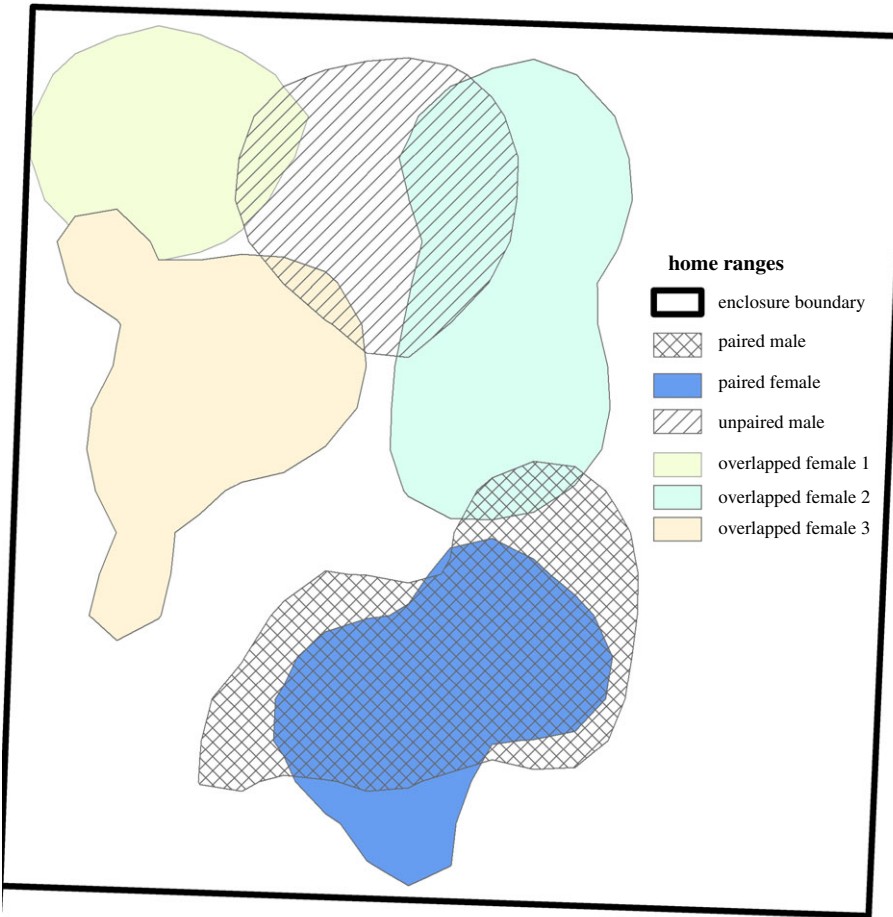

**Figure 5.** A sample of home ranges from one enclosure. Home ranges are 75% kernel utilization distributions from radio-tracking locations gathered over a four-week period. Here the difference in home ranges of a monogamous pair two lowest kernels and an unpaired male are shown. Three indices were determined from the home ranges: home range size, the area covered by the kernel square-metres; maximum overlap, the maximum proportion of a male's home range overlapped by a female's home range; and the proportion of females overlapped, which is the number of female home ranges a male's kernel overlaps, however slightly. (Online version in colour.)

Ugglan multiple-capture traps in each enclosure, arranged in a 5 × 5 grid, with approximately 5 m separating each trap. Traps were set and checked three evenings and two mornings for a total of five trap checks per week.

### (i) Social monogamy

Eight indices of social monogamy were assessed (table 1). Three were calculated from nest use data. If a male was trapped or found via radio transmitter nest-checks ≥75% of the time at one specific nest during a nest residency period (three periods based on trapping regimes: weeks 6–8, 9–11, 12–14), he was considered a resident of that nest and assigned a residency score of 1. Males that were detected less than 75% of the time at one nest during a nest residency period were given a score of 0. If a male was a resident at a nest with only one resident female (residency determined as for males), he was given a social monogamy score of 1. Males residing at nests with no females, more than one resident female, other resident males or that were not residents were assigned a social monogamy score of 0 [29]. We also used the same nest-trapping and radio nest-checking data to calculate a pair-wise half-weight AI [54] for every possible founding male-female combination in an enclosure using R (R Core Development Team 2017) as well as calculating a relative AI for each founding male following [29,30,55]. The relative AI indicates the strength of a male's association with the female he associated with the most compared to associations with other females.

Grid trapping (three periods: weeks 8, 11, 14–15) was used to calculate social monogamy based on spatial overlap; males were considered to overlap an adult female if both the male and

female were trapped at the same trap location even if captures were not simultaneous. The adult female that shared the most total trap locations with a male was considered the female he overlapped with most and the amount of overlap was determined by the number of sites at which the male and female were captured divided by the total number of sites where the male was captured during each grid-trapping period. The proportion of adult females that a male's home range overlapped was calculated as the total number of females a male shared a trap location with divided by the total number of adult females in the enclosure during each grid-trapping period.

While frequently used, trapping results are a relatively gross measure of home range location and size so we also used GPS points from radio-tracking to create 75% kernel utilization distributions for each vole [56] for three 4–week periods: weeks 3–6, 7–10 and 11–14. We excluded the first two weeks during which voles established home ranges and pairs and because voles that died during this time were replaced; the last week of the study was excluded because many of the vole's radio-collars had stopped functioning. For kernel utilization distributions, we used 75% probability contours, a smoothing parameter (h) of 2, a grid of 100 and extent of 2 for each enclosure using the adehabitatHR [57] package in R for calculating male home range size. Commonly used kernel parameters (e.g. 95% contours, reference bandwidth smoothing) overestimated vole's space use, while the selected parameters provided biologically sensible home ranges for the input relocations while not limiting the kernel to the smaller 50% cores, which are highly likely to miss female overlap [58]. The 'HR' method in adehabitatHR [59] was used to determine the maximum overlap a male shared with an adult female and

the proportion of adult females in the enclosure he overlapped for each period (figure 5).

### (ii) Parentage

Founders and offspring that were trapped, were genotyped at six polymorphic loci previously validated for parentage analyses in prairie voles [29,60]. We extracted DNA from tissue samples using DNeasy extraction kits (Qiagen, Valencia, CA, USA) and then amplified the microsatellites using polymerase chain reaction [60]. Cervus 3.0 [61,62] was used to assign parents with 95% confidence. We determined the number of females with which a male sired offspring and estimated the conception dates of offspring. Males that sired offspring with only one female during the entire study were classified as genetically monogamous and assigned a 1, while males that sired offspring with more than one female during the study were considered to not be genetically monogamous and assigned a 0.

### (iii) Tissue collection and staining

Upon completion of the study, surviving founding males were euthanized and their brains were collected and fixed in 4% paraformaldehyde. Free-floating sections were sectioned at 40 µm on a freezing sliding microtome. Due to failure of commercially available ERα primary antibodies, inhibition was assessed in two ways. Immunohistochemistry was used to visualize cells expressing GFP immunoreactivity in ERα RNAi transfected males using the anti-GFP primary antibody (Abcam, Cambridge, UK, ab290; diluted 1 : 1000; figure 2a), which does not contain intronic sequences or show cross-binding with any other transcripts of closely related receptors, and the expression or lack of ERα mRNA was visualized using RNAscope (HiPlex ADC, figure 2b–d).

### (iv) Statistical analyses

We analysed each index of social monogamy using mixed-effects models. For each of these models, we included the appropriate behavioural index as the response variable and the treatment (control or RNAi males) as the fixed effect of interest. We first found the combination of random effects with the lowest AIC for each response variable; individual vole IDs were included as a random effect in every model with enclosure and period as the other possible random effects. Survival was included as a fixed effect for analyses of total offspring. For binary responses (such as residency scores), we used generalized linear mixed models with binomial family distributions and the previously mentioned fixed and random effects. Every analysis had a full model with treatment (the fixed effect of interest), the selected random effects and interactions, plus another 'null' model that did not include the treatment. We compared these two models via a likelihood ratio test using the base 'anova' function to determine whether the addition of treatment significantly improved model fit. All analyses were performed in R v. 3.6.1 (R Core Development Team 2019) with the lme4 package. Although this approach entails multiple comparisons between treatments, we present the true p-values using $\alpha = 0.05$ for each analysis rather than a correction factor due to the overly conservative nature of such corrections for ecological experiments [63–65]. At the end of the study, 51.6% of the founding males and 42.2% of the founding females had survived. There was no difference in mortality between types of males. Because only some males survived until the end of the study and subsequently had their brains collected to verify ERα inhibition in the MeA, we conducted two sets of analyses as described in the above results.

Ethics. All procedures were approved by the Miami University IACUC and followed guidelines for using wild mammals in research from the American Society of Mammalogists and in accordance with the National Institutes Health Guide for the Care and Use of Laboratory Animals.

Data accessibility. All data and code used for the analyses are available via the Dryad Digital Repository: https://doi.org/10.5061/dryad.z08kprrc3 [66].

The data are provided in electronic supplementary material [67].

Authors' contributions. C.T.L.: formal analysis, investigation, writing-original draft, writing-review and editing; J.B.L.: data curation, methodology; A.N.P.: investigation, methodology; S.A.C.: formal analysis, methodology, validation; B.K.: funding acquisition, methodology, project administration, supervision, writing—original draft, writing—review and editing; B.S.C.: conceptualization, funding acquisition, investigation, project administration, writing-original draft, writing—review and editing; N.G.S.: funding acquisition, project administration, supervision, writing—original draft.

All authors gave final approval for publication and agreed to be held accountable for the work performed therein.

Competing interests. We declare we have no competing interests.

Funding. This project was funded by the National Institutes of Health grant (grant no. 1R15HD075222-01A1) awarded to N.G.S., B.K. and B.S.C., and through an American Society of Mammalogists Grant-in-Aid of Research, American Museum of Natural History Theodore Roosevelt Memorial Fund, Sigma Xi Grant-in-Aid of Research to C.T.L., NSF 1826745 'Louis Stokes STEM Pathways and Research Alliance: University of Texas System LSAMP' in support of S.A.C., Larry P. Jones Distinguished Professor Endowment supporting B.S.C. and Miami University's Biology Department.

Acknowledgements. We wish to thank the staff at the Miami University's Ecology Research Center for the assistance with the enclosure study and the many undergraduates who helped.

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
