## [Peer Review File · Proceedings of the Royal Society B: Biological Sciences]

Review History

RSPB-2021-0318.R0 (Original submission)

Review form: Reviewer 1

Recommendation

Accept with minor revision (please list in comments)

Scientific importance: Is the manuscript an original and important contribution to its field?

Excellent

General interest: Is the paper of sufficient general interest?

Excellent

Quality of the paper: Is the overall quality of the paper suitable?

Excellent

Is the length of the paper justified?

Yes

Should the paper be seen by a specialist statistical reviewer?

No

Do you have any concerns about statistical analyses in this paper? If so, please specify them explicitly in your report.

No

It is a condition of publication that authors make their supporting data, code and materials available - either as supplementary material or hosted in an external repository. Please rate, if applicable, the supporting data on the following criteria.

Is it accessible?

No

Is it clear?

N/A

Is it adequate?

N/A

Do you have any ethical concerns with this paper?

No

Comments to the Author

Overall, this is an interesting and creative study that provides compelling evidence for the role of estrogen receptors in social monogamy. The use of an ecologically relevant, field setting to test these behaviors is admirable, given the challenges associated with integrating field and technically challenging bench science (i.e., using adeno-associated viral vectors to knockdown gene expression) in one study. These types of studies are not often done (particularly in mammals) but are critical for our understanding of how neurobiological differences translate to actual behavior in natural systems.

There are a few areas where the clarity of the manuscript could be improved, or where some additional information is needed, which are outlined below in the order in which they appear in the manuscript, not importance.

1) The use of the term “overexpression” to describe the control animals is somewhat confusing, because overexpression implies experimental manipulation to increase ER α levels above naturally occurring levels. Describing the KI males as a male type with “high ER levels and low measures of social monogamy” or equivalent would be more clear.

INTRODUCTION

2) Line 88: “This prediction is based upon a hypothesized relationship and empirical findings” This phrase is necessary as one would assume this to be the case.

3) Line 91: should be “nonapeptides”

4) Line 96: should be “are caused by”

5) The field evidence for ER α 's role in male offspring provisioning and territoriality in sparrows would be relevant here, or in the discussion, to generalize findings to other vertebrate groups: Horton, B. M., Hudson, W. H., Ortlund, E. A., Shirk, S., Thomas, J. W., Young, E. R., ... & Maney, D. L. (2014). Estrogen receptor α polymorphism in a species with alternative behavioral phenotypes. *Proceedings of the National Academy of Sciences*, 111, 1443-1448.

METHODS

6) What is the total animal number per enclosure/enclosure density, or number of enclosures

used? The potential effect of density on social monogamy is referenced a few times, so this is important information to include to allow comparison with future studies or allow aspects of the work to be replicated.

7) Lines 320-323 : “Two analyses were conducted, one comparing data from only confirmed MeA knockdown males (n=9) against control males (n = 29) and a more conservative comparison of data from all ER α RNAi-transfected founding males (n=32), regardless of whether MeA knockdown could be confirmed, against control males.”

This should be moved up to before the statistics description, so the reader is not left wondering throughout about sample size and which groups are being analyzed.

8) “We first found the combination of random effects with the lowest AIC for each response variable; individual vole ID’s were included as a random effect in every model with enclosure and period as the other possible random effects.”

Given the sample size in the smaller group, up to 3 random effects is a lot, raising the question if the random effects are truly being fit by the models. Though the authors took a conservative approach by using both groups (confirmed manipulated and all manipulated) and found similar results in both analyses.

9) Corrected AIC (AICc) would be better to use for this given the sample size/parameters, or in future similar analyses.

Resources: Harrison, X. A., Donaldson, L., Correa-Cano, M. E., Evans, J., Fisher, D. N., Goodwin, C. E., ... & Inger, R. (2018). A brief introduction to mixed effects modelling and multi-model inference in ecology. *PeerJ*, 6, e4794.

<https://cran.r-project.org/web/packages/AICcmodavg/AICcmodavg.pdf>

RESULTS

10) The methods state that different models were chosen (certain random effects included or not included) for different response variables based on model AIC. However, in the results or elsewhere, it is not clear which parameters were actually used in the final analysis for each response variable. A table or supplementary table with the models chosen and output would be useful here. Or adding a column to table 1 on the parameters used for each of the response variables.

11) Is there a 20x image in Fig 2b?

DISCUSSION

12) Lines 164-166: “The robust effects following the manipulation of ER α expression contrast with the consistent lack of direct translation associated with other mechanisms, e.g., *avrp1a* microsatellite length, to predictable outcomes in field studies [27,29,30,45].”

The failure of AVP mechanisms in directly translating to behavior is mentioned in the introduction and here, but not elaborated on. It would be beneficial to know what exactly those cited studies found (in brief) and the shortcomings, to better compare them to the results here.

13) Lines 188-190: “First, even though the sample size of surviving males in which ER α was verified to be successfully knocked down was limited, decreasing the probability of finding a significant effect, there were still numerous differences, with five out of eight social monogamy indices being significantly different.”

Yes, but low sample size also increases the likelihood that significant findings don’t represent true effects, because it is more likely that your sample is not truly representative of the

population. It also inflates effect size. Thus implying that limited sample size makes the significant findings more robust, is not really true and should be avoided. (see: Button, K. S., Ioannidis, J. P., Mokrysz, C., Nosek, B. A., Flint, J., Robinson, E. S., & Munafò, M. R. (2013). Power failure: why small sample size undermines the reliability of neuroscience. *Nature reviews neuroscience*, 14, 365-376.)

Review form: Reviewer 2

Recommendation

Accept with minor revision (please list in comments)

Scientific importance: Is the manuscript an original and important contribution to its field?

Excellent

General interest: Is the paper of sufficient general interest?

Excellent

Quality of the paper: Is the overall quality of the paper suitable?

Acceptable

Is the length of the paper justified?

Yes

Should the paper be seen by a specialist statistical reviewer?

No

Do you have any concerns about statistical analyses in this paper? If so, please specify them explicitly in your report.

Yes

It is a condition of publication that authors make their supporting data, code and materials available - either as supplementary material or hosted in an external repository. Please rate, if applicable, the supporting data on the following criteria.

Is it accessible?

No

Is it clear?

No

Is it adequate?

No

Do you have any ethical concerns with this paper?

No

Comments to the Author

Very exciting and challenging study. The authors use a genetic cross to generate prairie voles that have high ERa expression in the MeA to test the role of ERa in social and genetic monogamy in a semi-naturalistic setting. Using gene knockdown, they convincingly demonstrate that expression of ERa in MeA contributes to social monogamy, confirming earlier lab-based studies. The manuscript would be improved by making a few small adjustments, outlined below. The introduction and discussion rely heavily on avpr1a and prairie vole literature when the authors

could discuss the large literature on the role of steroid hormones and their expression in prosocial behavior across vertebrates.

Line 78. The role of AVP in social monogamy has been translated to *Peromyscus* mice, see Hopi Hoekstra and Cathy Marler's work on the role of AVP

Line 85. Evidence that water balance contributes to social behaviors via AVP?

Line 97-99. It is worth reviewing some of the work on the effects of estrogen receptor alpha in species that show pair bonding and biparental care (Trainor, Marler, Wynne-Edwards, and Maney's work)

Line 128. Show survival data in supplemental materials and consider moving the section on survival to the methods rather than the beginning of the results to not detract from the most important findings

Line 161. This is not the first study to show a mechanism of social monogamy also affects behavior in the wild. Consider T manipulations in California mice, as an example.

Line 258-259. The same size will be small, but did the knockdown effect non-monogamous males by changing the number of females associated with a male or the size of the territory?

Line 278. Dissociating the persistence and strength of the pair bond from pair bond formation is challenging. It is rare to have a naturalistic dataset that could shed light on mechanisms of pair bond formation. Was there a difference between knockdown and control males in the rate at which pairs and territories were established? Or rates of "divorce"?

Line 296. How many males that were considered to be not genetically monogamous changed their primary partner during the trapping period?

Line 305. What was the sequence of the ERa probe for RNAscope, was it intron spanning, and did it have any substantial sequence similarity to closely related genes such as ERb or other steroid receptors?

Line 306. Authors need to report the packages, versions of packages, R or RStudio, and versions of R that were used to analyze the data and cite the relevant sources. Need to state the package used to perform an ANOVA to test the model fits, and cite sources validating why the ANOVA "null" analysis and AIC model selection are appropriate. I assume that the data that are presented are Wald chi-squared tests (glm's do not automatically give chi-squared values) but that is not stated in the statistics section. The initial vs final models should be reported in the supplemental information with the p-value and AIC used to justify the final model. It would be helpful to others to report code used to analyze the data.

Figure 2. The data showing the extent of the knockdown versus controls should be presented. Is there an effect of bilateral vs unilateral knockdown? How many animals had bilateral knockdown vs. unilateral knockdown?

Figures 3,4. The authors should show individual data points in all figures. All data should be supplied in the supplemental materials and/or deposited on Dryad. Indicate which panels of Figure 4 show inconsistent results when compared to Figure 3.

Decision letter (RSPB-2021-0318.R0)

09-Apr-2021

Dear Dr Cushing:

Your manuscript has now been peer reviewed and the reviews have been assessed by an Associate Editor. The reviewers' comments (not including confidential comments to the Editor) and the comments from the Associate Editor are included at the end of this email for your reference. As you will see, the reviewers and the Editors have raised some concerns with your manuscript and we would like to invite you to revise your manuscript to address them.

Research ethics:

Use of animals and field studies:

It is a condition of publication that you make available the data and research materials supporting the results in the article. Please see our Data Sharing Policies (<https://royalsociety.org/journals/authors/author-guidelines/#data>). Datasets should be deposited in an appropriate publicly available repository and details of the associated accession number, link or DOI to the datasets must be included in the Data Accessibility section of the

article (<https://royalsociety.org/journals/ethics-policies/data-sharing-mining/>). Reference(s) to datasets should also be included in the reference list of the article with DOIs (where available).

Please submit a copy of your revised paper within three weeks. If we do not hear from you within this time your manuscript will be rejected. If you are unable to meet this deadline please let us know as soon as possible, as we may be able to grant a short extension.

Best wishes,
Dr Sasha Dall
mailto: proceedingsb@royalsociety.org

Associate Editor

Board Member: 1

Comments to Author:

We have now heard from two experts, both of whom are enthusiastic about your study. They do have some suggestions for improving that clarity of your manuscript before we move forward. Some details about methods and statistical procedures are required.

Reviewer(s)' Comments to Author:

Referee: 1

Comments to the Author(s)

Overall, this is an interesting and creative study that provides compelling evidence for the role of estrogen receptors in social monogamy. The use of an ecologically relevant, field setting to test these behaviors is admirable, given the challenges associated with integrating field and technically challenging bench science (i.e., using adeno-associated viral vectors to knockdown gene expression) in one study. These types of studies are not often done (particularly in

mammals) but are critical for our understanding of how neurobiological differences translate to actual behavior in natural systems.

There are a few areas where the clarity of the manuscript could be improved, or where some additional information is needed, which are outlined below in the order in which they appear in the manuscript, not importance.

1) The use of the term “overexpression” to describe the control animals is somewhat confusing, because overexpression implies experimental manipulation to increase ER α levels above naturally occurring levels. Describing the KI males as a male type with “high ER levels and low measures of social monogamy” or equivalent would be more clear.

INTRODUCTION

2) Line 88: “This prediction is based upon a hypothesized relationship and empirical findings” This phrase is necessary as one would assume this to be the case.

3) Line 91: should be “nonapeptides”

4) Line 96: should be “are caused by”

5) The field evidence for ER α 's role in male offspring provisioning and territoriality in sparrows would be relevant here, or in the discussion, to generalize findings to other vertebrate groups: Horton, B. M., Hudson, W. H., Ortlund, E. A., Shirk, S., Thomas, J. W., Young, E. R., ... & Maney, D. L. (2014). Estrogen receptor α polymorphism in a species with alternative behavioral phenotypes. *Proceedings of the National Academy of Sciences*, 111, 1443-1448.

METHODS

6) What is the total animal number per enclosure/enclosure density, or number of enclosures used? The potential effect of density on social monogamy is referenced a few times, so this is important information to include to allow comparison with future studies or allow aspects of the work to be replicated.

7) Lines 320-323 : “Two analyses were conducted, one comparing data from only confirmed MeA knockdown males (n=9) against control males (n = 29) and a more conservative comparison of data from all ER α RNAi-transfected founding males (n=32), regardless of whether MeA knockdown could be confirmed, against control males.”

This should be moved up to before the statistics description, so the reader is not left wondering throughout about sample size and which groups are being analyzed.

8) “We first found the combination of random effects with the lowest AIC for each response variable; individual vole ID's were included as a random effect in every model with enclosure and period as the other possible random effects.”

Given the sample size in the smaller group, up to 3 random effects is a lot, raising the question if the random effects are truly being fit by the models. Though the authors took a conservative approach by using both groups (confirmed manipulated and all manipulated) and found similar results in both analyses.

9) Corrected AIC (AICc) would be better to use for this given the sample size/parameters, or in future similar analyses.

Resources: Harrison, X. A., Donaldson, L., Correa-Cano, M. E., Evans, J., Fisher, D. N., Goodwin, C. E., ... & Inger, R. (2018). A brief introduction to mixed effects modelling and multi-model inference in ecology. *PeerJ*, 6, e4794.

<https://cran.r-project.org/web/packages/AICcmodavg/AICcmodavg.pdf>

RESULTS

10) The methods state that different models were chosen (certain random effects included or not included) for different response variables based on model AIC. However, in the results or elsewhere, it is not clear which parameters were actually used in the final analysis for each response variable. A table or supplementary table with the models chosen and output would be useful here. Or adding a column to table 1 on the parameters used for each of the response variables.

11) Is there a 20x image in Fig 2b?

DISCUSSION

12) Lines 164-166: "The robust effects following the manipulation of ER α expression contrast with the consistent lack of direct translation associated with other mechanisms, e.g., avrp1a microsatellite length, to predictable outcomes in field studies [27,29,30,45]."

The failure of AVP mechanisms in directly translating to behavior is mentioned in the introduction and here, but not elaborated on. It would be beneficial to know what exactly those cited studies found (in brief) and the shortcomings, to better compare them to the results here.

13) Lines 188-190: "First, even though the sample size of surviving males in which ER α was verified to be successfully knocked down was limited, decreasing the probability of finding a significant effect, there were still numerous differences, with five out of eight social monogamy indices being significantly different."

Yes, but low sample size also increases the likelihood that significant findings don't represent true effects, because it is more likely that your sample is not truly representative of the population. It also inflates effect size. Thus implying that limited sample size makes the significant findings more robust, is not really true and should be avoided. (see: Button, K. S., Ioannidis, J. P., Mokrysz, C., Nosek, B. A., Flint, J., Robinson, E. S., & Munafò, M. R. (2013). Power failure: why small sample size undermines the reliability of neuroscience. *Nature reviews neuroscience*, 14, 365-376.)

Referee: 2

Comments to the Author(s)

Very exciting and challenging study. The authors use a genetic cross to generate prairie voles that have high ER α expression in the MeA to test the role of ER α in social and genetic monogamy in a semi-naturalistic setting. Using gene knockdown, they convincingly demonstrate that expression of ER α in MeA contributes to social monogamy, confirming earlier lab-based studies. The manuscript would be improved by making a few small adjustments, outlined below. The introduction and discussion rely heavily on avrp1a and prairie vole literature when the authors could discuss the large literature on the role of steroid hormones and their expression in prosocial behavior across vertebrates.

Line 78. The role of AVP in social monogamy has been translated to *Peromyscus* mice, see Hopi Hoekstra and Cathy Marler's work on the role of AVP

Line 85. Evidence that water balance contributes to social behaviors via AVP?

Line 97-99. It is worth reviewing some of the work on the effects of estrogen receptor alpha in species that show pair bonding and biparental care (Trainor, Marler, Wynne-Edwards, and Maney's work)

Line 128. Show survival data in supplemental materials and consider moving the section on survival to the methods rather than the beginning of the results to not detract from the most important findings

Line 161. This is not the first study to show a mechanism of social monogamy also affects behavior in the wild. Consider T manipulations in California mice, as an example.

Line 258-259. The same size will be small, but did the knockdown effect non-monogamous males by changing the number of females associated with a male or the size of the territory?

Line 278. Dissociating the persistence and strength of the pair bond from pair bond formation is challenging. It is rare to have a naturalistic dataset that could shed light on mechanisms of pair bond formation. Was there a difference between knockdown and control males in the rate at which pairs and territories were established? Or rates of “divorce”?

Line 296. How many males that were considered to be not genetically monogamous changed their primary partner during the trapping period?

Line 305. What was the sequence of the ERa probe for RNAscope, was it intron spanning, and did it have any substantial sequence similarity to closely related genes such as ERb or other steroid receptors?

Line 306. Authors need to report the packages, versions of packages, R or RStudio, and versions of R that were used to analyze the data and cite the relevant sources. Need to state the package used to perform an ANOVA to test the model fits, and cite sources validating why the ANOVA “null” analysis and AIC model selection are appropriate. I assume that the data that are presented are Wald chi-squared tests (glmm’s do not automatically give chi-squared values) but that is not stated in the statistics section. The initial vs final models should be reported in the supplemental information with the p-value and AIC used to justify the final model. It would be helpful to others to report code used to analyze the data.

Figure 2. The data showing the extent of the knockdown versus controls should be presented. Is there an effect of bilateral vs unilateral knockdown? How many animals had bilateral knockdown vs. unilateral knockdown?

Figures 3,4. The authors should show individual data points in all figures. All data should be supplied in the supplemental materials and/or deposited on Dryad. Indicate which panels of Figure 4 show inconsistent results when compared to Figure 3.

Author's Response to Decision Letter for (RSPB-2021-0318.R0)

See Appendix A.

Decision letter (RSPB-2021-0318.R1)

25-Jun-2021

Dear Dr Cushing

I am pleased to inform you that your manuscript RSPB-2021-0318.R1 entitled "Medial amygdala ER α expression influences monogamous behavior of male prairie voles in the field" has been accepted for publication in Proceedings B.

The AE has recommended publication, but also suggests some minor revisions to your manuscript. Therefore, I invite you to respond to the comments and revise your manuscript. Because the schedule for publication is very tight, it is a condition of publication that you submit the revised version of your manuscript within 7 days. If you do not think you will be able to meet this date please let us know.

[http://datadryad.org/submit?journalID=RSPB&manu=\(Document not available\)](http://datadryad.org/submit?journalID=RSPB&manu=(Document%20not%20available)) which will take you to your unique entry in the Dryad repository. If you have already submitted your data to dryad you can make any necessary revisions to your dataset by following the above link. Please see <https://royalsociety.org/journals/ethics-policies/data-sharing-mining/> for more details.

Sincerely,
Dr Sasha Dall
Editor, Proceedings B
<mailto:proceedingsb@royalsociety.org>

Associate Editor:
Board Member
Comments to Author:

I found your response to the reviews persuasive. There still appears to be a problem with Figure 2b at least to my eye. As Reviewer 1 says, the 20x image appears to be missing or is not evident. Otherwise the revision looks great.

Author's Response to Decision Letter for (RSPB-2021-0318.R1)

See Appendix B.

Decision letter (RSPB-2021-0318.R2)

12-Jul-2021

Dear Dr Cushing

I am pleased to inform you that your manuscript entitled "Medial amygdala ER α expression influences monogamous behavior of male prairie voles in the field" has been accepted for publication in Proceedings B.

Data Accessibility section

Open Access

Paper charges

Sincerely,

Dr Sasha Dall

Associate Editor:

Board Member

Comments to Author:

Thanks for fixing the figure. It all looks good. Congrats on what looks to be a fascinating study.

Appendix A

Reviewer(s)' Comments to Author:

Referee: 1

Comments to the Author(s)

Overall, this is an interesting and creative study that provides compelling evidence for the role of estrogen receptors in social monogamy. The use of an ecologically relevant, field setting to test these behaviors is admirable, given the challenges associated with integrating field and technically challenging bench science (i.e., using adeno-associated viral vectors to knockdown gene expression) in one study. These types of studies are not often done (particularly in mammals) but are critical for our understanding of how neurobiological differences translate to actual behavior in natural systems.

There are a few areas where the clarity of the manuscript could be improved, or where some additional information is needed, which are outlined below in the order in which they appear in the manuscript, not importance.

1) The use of the term “overexpression” to describe the control animals is somewhat confusing, because overexpression implies experimental manipulation to increase ER α levels above naturally occurring levels. Describing the KI males as a male type with “high ER levels and low measures of social monogamy” or equivalent would be more clear.

We have changed this wording throughout the manuscript; see L26, L125, L239

INTRODUCTION

2) Line 88: “This prediction is based upon a hypothesized relationship and empirical findings” This phrase is necessary as one would assume this to be the case.

We have removed this line as suggested.

3) Line 91: should be “nonapeptides”

Changed, L98.

4) Line 96: should be “are caused by”

Changed, L103

5) The field evidence for ER α 's role in male offspring provisioning and territoriality in sparrows would be relevant here, or in the discussion, to generalize findings to other vertebrate groups: Horton, B. M., Hudson, W. H., Ortlund, E. A., Shirk, S., Thomas, J. W., Young, E. R., ... & Maney, D. L. (2014). Estrogen receptor α polymorphism in a species with alternative behavioral phenotypes. *Proceedings of the National Academy of Sciences*, 111, 1443-1448.

We thank the reviewer for this suggestion; we had cited this paper in previous drafts and have now added a sentence in the introduction referencing it, L117-118.

METHODS

6) What is the total animal number per enclosure/enclosure density, or number of enclosures used? The potential effect of density on social monogamy is referenced a few times, so this is important information to include to allow comparison with future studies or allow aspects of the work to be replicated.

We have added in this information, see L244-45

7) Lines 329-332 : “Two analyses were conducted, one comparing data from only confirmed MeA knockdown males (n=9) against control males (n = 29) and a more conservative comparison of data from all ER α RNAi-transfected founding males (n=32), regardless of whether MeA knockdown could be confirmed, against control males.”

This should be moved up to before the statistics description, so the reader is not left wondering throughout about sample size and which groups are being analyzed.

We have moved this into the beginning of the results and slightly modified the surrounding text, L148-51.

8) “We first found the combination of random effects with the lowest AIC for each response variable; individual vole ID’s were included as a random effect in every model with enclosure and period as the other possible random effects.”

Given the sample size in the smaller group, up to 3 random effects is a lot, raising the question if the random effects are truly being fit by the models. Though the authors took a conservative approach by using both groups (confirmed manipulated and all manipulated) and found similar results in both analyses.

We understand the concern of the reviewer here and note that though the samples size of individuals is small as indicated, our analyses consisted of 49-89 total individual data points, as we had up to three repeated measurements for each individual vole across the different periods of the study. We also note that none of the analyses included all 3 random effects- vole ID was always included due to the repeated measures, enclosure was never included, and the period was included only in some of the models. See Supplemental Tables 2-5, added as part of the supplementary material.

9) Corrected AIC (AICc) would be better to use for this given the sample size/parameters, or in future similar analyses.

Resources: Harrison, X. A., Donaldson, L., Correa-Cano, M. E., Evans, J., Fisher, D. N., Goodwin, C. E., ... & Inger, R. (2018). A brief introduction to mixed effects modelling and multi-model inference in ecology. PeerJ, 6, e4794.

We appreciate the insight here, and note again our overall number of data points while acknowledging this limitation. Because our use of AIC was only tangential in our analyses (for selecting random effects, not comparing our actual effects of interest) we expect that taking this into account would have little to no impact on our findings.

RESULTS

10) The methods state that different models were chosen (certain random effects included or not included) for different response variables based on model AIC. However, in the results or elsewhere, it is not clear which parameters were actually used in the final analysis for each response variable. A table or supplementary table with the models chosen and output would be useful here. Or adding a column to table 1 on the parameters used for each of the response variables.

We have now added multiple tables to the new supplemental material, including two tables displaying the statistical information for all of the models and analyses and two tables showing AIC values for the differing models. These tables include information on what random effects were used in each model.

11) Is there a 20x image in Fig 2b?

Unless there was something that didn't translate in the submission/some mistake, both B and D should have a 20x image. There are three images in B and D marked.

DISCUSSION

12) Lines 173-175: "The robust effects following the manipulation of ER α expression contrast with the consistent lack of direct translation associated with other mechanisms, e.g., avrp1a microsatellite length, to predictable outcomes in field studies [27,29,30,45]."

The failure of AVP mechanisms in directly translating to behavior is mentioned in the introduction and here, but not elaborated on. It would be beneficial to know what exactly those cited studies found (in brief) and the shortcomings, to better compare them to the results here.

We have added a sentence in the introduction expanding on these previous findings, L77-88.

13) Lines 197-199: "First, even though the sample size of surviving males in which ER α was verified to be successfully knocked down was limited, decreasing the probability of finding a significant effect, there were still numerous differences, with five out of eight social monogamy indices being significantly different."

Yes, but low sample size also increases the likelihood that significant findings don't represent true effects, because it is more likely that your sample is not truly representative of the population. It also inflates effect size. Thus implying that limited sample size makes the significant findings more robust, is not really true and should be avoided. (see: Button, K. S., Ioannidis, J. P., Mokrysz, C., Nosek, B. A., Flint, J., Robinson, E. S., & Munafò, M. R. (2013). Power failure: why small sample size undermines the reliability of neuroscience. *Nature reviews neuroscience*, 14, 365-376.)

We thank the author for pointing this- we recognize our error and have corrected this section extensively, see L210-215.

Referee: 2

Comments to the Author(s)

Very exciting and challenging study. The authors use a genetic cross to generate prairie voles that have high ER α expression in the MeA to test the role of ER α in social and genetic monogamy in a semi-naturalistic setting. Using gene knockdown, they convincingly demonstrate that expression of ER α in MeA contributes to social monogamy, confirming earlier lab-based studies. The manuscript would be improved by making a few small adjustments, outlined below. The introduction and discussion rely heavily on avpr1a and prairie vole literature when the authors could discuss the large literature on the role of steroid hormones and their expression in prosocial behavior across vertebrates.

Line 78. The role of AVP in social monogamy has been translated to *Peromyscus* mice, see Hope Hoekstra and Cathy Marler's work on the role of AVP

We are unaware of a direct reference specifically linking AVP with social monogamy. There are studies demonstrating a role of AVP in male behavior, especially directed toward pups, a characteristic of social monogamy, but not pair bonding. Cushing (Plos One) showed a relationship between proportional expression AVP and ER α , but not a direct relationship with only AVP expression, and Fink et al specifically demonstrated that there was not a relationship between the V1a receptor and mating strategy. The relationship between microsatellite length of the receptor, high levels of AVP, and social monogamy in voles has not necessarily been the same in other species, so since the role of AVP is not the same it would be misleading to express otherwise. Also early work did not support the same relationship as Meredith et al (1999) reported parental behavior or males was not associated with AVP.

Line 85. Evidence that water balance contributes to social behaviors via AVP?

We have added a citation here, L93

Line 97-99. It is worth reviewing some of the work on the effects of estrogen receptor alpha in species that show pair bonding and biparental care (Trainor, Marler, Wynne-Edwards, and Maney's work)

We have added in some more on this (L114-118) but for the sake of page and word limitations have kept it brief; we can expand further if the reviewer or editor feels it important.

Line 128. Show survival data in supplemental materials and consider moving the section on survival to the methods rather than the beginning of the results to not detract from the most important findings

We have included the statistics on the survival comparison in the new tables we've added in the new supplementary material, and moved the survival information into the

results as suggested, see L148.

Line 161. This is not the first study to show a mechanism of social monogamy also affects behavior in the wild. Consider T manipulations in California mice, as an example.

*We would disagree- the studies we are aware of and *think* the reviewer is referencing deals with vocalizations and place preferences, and not social monogamy or pair-bonding per se.*

Line 267-268. The same size will be small, but did the knockdown effect non-monogamous males by changing the number of females associated with a male or the size of the territory?

We are uncertain exactly what is being asked here- RNAi knockdown males did have smaller territories and overlapped more females, as shown in the data.

Line 287. Dissociating the persistence and strength of the pair bond from pair bond formation is challenging. It is rare to have a naturalistic dataset that could shed light on mechanisms of pair bond formation. Was there a difference between knockdown and control males in the rate at which pairs and territories were established? Or rates of “divorce”?

We did not examine the rate at which pairs and territories were established for a few reasons. First, we did not trap during the first six weeks of the experiment and though we collected radiotracking data during this time, it would be difficult to determine precise establishment of bonds with the radiotracking data we have. This data worked very well for examining home ranges across four-week periods but smaller timespans would result in limited data points to calculate anything meaningful.

As suggested, we conducted an analysis aimed at examining ‘divorce’ by looking at the number of primary social partners each male had over the course of the study, excluding changes due to death of their partner. RNAi and sham males did not differ in this, with only 1/9 RNAi males having over 1 social partner and only 7/29 sham voles. Even when examining only those males that survived the entire study or examining all RNAi voles and all sham voles (while controlling for length of survival) there were no differences. As these analyses did not find anything new, we did not add them to the manuscript but are willing to if desired.

Line 305. How many males that were considered to be not genetically monogamous changed their primary partner during the trapping period?

The way our data was collected and analyzed does not allow us to examine if voles changed partners during any specific period, as males could only be assigned to a partner based on the data collected in that period. However, to address this as best we could we conducted a few analyses that we hope get at the question posed. We compared the number of social partners (calculated as done for the ‘divorce’ analysis suggested above) between RNAi and control males that were not genetically monogamous across the entire study but found no differences- 3/4 RNAi males had only 1 social partner, and 8/12 control males had only 1 social partner. As with the last question we did not this into the manuscript as it did not add anything new and, while

interesting, seemed tangential to our main questions, but are more than willing to add it in if requested.

Line 314. What was the sequence of the ERa probe for RNAscope, was it intron spanning, and did it have any substantial sequence similarity to closely related genes such as ERb or other steroid receptors?

The probe for RNAscope was not intron spanning and is highly unlikely to have any cross-binding with other steroid receptors; we have added in a line to the manuscript (L328-329) to reflect this. Here is the direct reply from the company regarding this: "We have used the sequence from accession number NM_001282251.1 to design the probe Mo-Esr1 (ERa, 20ZZ targeting from 936-1813). As you can see, this sequence does not contain any intronic sequences. Also, we do not detect substantial sequence homology between Esr1 and other closely related steroid receptors (see BLAST results here: B2N5WHM3013 Search expires on 05-30 04:38 am) in Microtus ochrogaster. Furthermore, I have verified that the probe Mo-Esr1 does not cross-react with Esr2 or any other closely related steroid receptor transcript."

Line 306. Authors need to report the packages, versions of packages, R or RStudio, and versions of R that were used to analyze the data and cite the relevant sources. Need to state the package used to perform an ANOVA to test the model fits, and cite sources validating why the ANOVA "null" analysis and AIC model selection are appropriate. I assume that the data that are presented are Wald chi-squared tests (glmm's do not automatically give chi-squared values) but that is not stated in the statistics section. The initial vs final models should be reported in the supplemental information with the p-value and AIC used to justify the final model. It would be helpful to others to report code used to analyze the data.

We have added the information regarding R, the packages, and analyses into the manuscript (L342-343)- note that we conducted likelihood ratio tests. We have also added tables with AIC and other information into the new supplementary material.

Figure 2. The data showing the extent of the knockdown versus controls should be presented. Is there an effect of bilateral vs unilateral knockdown? How many animals had bilateral knockdown vs. unilateral knockdown?

We are unsure what is meant by the 'extent' of knockdown, but 6/9 'hit' voles had knockdown on both sides with the other three being hits unilateral (this has been added to the results)- and we note that hits result in complete knock-out of ERa at the hit site. Previous studies using this RNA knocked in the MeA indicated that there was no significant difference between unilateral and bilateral knockdown (see Cushing et al 2008, and Sheehan et al 2001). While we cannot rule out possible effects based upon the sample size, but the limited sample size precludes the power to detect difference if it did exist- and so we have left the analyses as they are. The size of the sample and the power of the results indicate that this was not a major factor, suggesting that even unilateral has a major effect on behavior.

Figures 3,4. The authors should show individual data points in all figures. All data should be supplied in the supplemental materials and/or deposited on Dryad. Indicate which panels of Figure 4 show inconsistent results when compared to Figure 3.

We understand the suggestion to add individual data points to the figure, but because individuals have repeated measures this would result in upwards of 80 data points on some of the figures and we do not think this would be beneficial for communicating the results in these figures. We have added symbology into Figure 4 to indicate which results differ in comparison to Figure 3. Also, as required upon submission, all of the data and code used to analyze the data is deposited in Dryad.

Appendix B

Dear Dr. Dall-

Response

We have revised the manuscript to meet the specifications for publication. Our manuscript should contain the all of the requisite information, including figure captions, and all tracked changes have been accepted. The figures are all be submitted as requested, and we have added a section regarding data accessibility following the ACKNOWLEDGEMENTS along with the citation to our dryad data (reference 66). Our supplementary material file is also prepared with slight modifications to the first page, and we added a sentence referencing our supplementary material on L182. We have included a media summary in our submission.

We have also made a few minor edits caught upon preparation on the following lines: L141-142, L145, L464, L467.

In response to the associate editor, we have fixed Figure 2 and the 20x image should be there now.